# Learning diverse causally emergent representations from time series data

**David McSharry**[*]
Department of Computing
Imperial College London
dm2223@ic.ac.uk

**Christos Kaplanis**[*]
Google DeepMind
kaplanis@google.com

**Fernando E. Rosas**
Sussex AI, University of Sussex
f.rosas@sussex.ac.uk

**Pedro A.M. Mediano**
Department of Computing
Imperial College London
p.mediano@imperial.ac.uk

## Abstract

Cognitive processes usually take place at a macroscopic scale in systems characterised by emergent properties that make the whole 'more than the sum of its parts.' While recent proposals have provided quantitative, information-theoretic metrics to detect emergence in time series data, it is often highly non-trivial to identify the relevant macroscopic variables a priori. In this paper we leverage recent advances in representation learning and differentiable information estimators to put forward a data-driven method to find variables with emergent properties. The proposed method successfully detects variables that exhibit emergent behaviour and recovers the ground-truth emergence values in a synthetic dataset. Furthermore, we show the method can be extended to learn multiple independent features, extracting a diverse set of emergent quantities. We finally show that a modified method scales to real experimental data from several brain activity datasets, paving the ground for future analyses uncovering the emergent structure of cognitive representations in biological and artificial intelligence systems.

## 1 Introduction

Cognitive processes usually take place in systems made of multiple interacting parts, e.g. neurons composing the nervous system of an organism. Importantly, cognitive processes themselves don't seem to take place at a 'microscopic' level of individual units, but at 'macroscopic' levels involving assemblies of several coordinated units [28]. Hence, when trying to unveil the inner workings of a — natural or artificial — cognitive system, it is crucial to be able to identify relevant macroscopic variables that best characterise the corresponding cognitive processes.

The identification of macroscopic variables has traditionally been driven by intuition and expert knowledge. For example, the investigation of collective behaviour in statistical physics is based on macroscopic variables known as 'order parameters,' which are typically identified heuristically and then used to describe phase transitions and other phenomena of interest [49]. Unfortunately, identifying relevant macroscopic variables is often more an art than a science, being heavily dependent on prior knowledge and expectations. Having automated procedures to identify relevant macroscopic variables of cognitive systems would open important avenues for investigating the inner workings of different cognitive architectures.

A promising approach to identify empirically useful macroscopic variables is provided by unsupervised representation learning [38, 22, 52]. For example, information maximisation has proven to be

38th Conference on Neural Information Processing Systems (NeurIPS 2024).

a powerful objective for learning representations within neural networks [23, 38]. In this paper we combine this approach with recent breakthroughs in our ability to formally characterise emergent phenomena [43, 34], which have proven to be not only theoretically sound but also empirically powerful [30, 40].

Building on this literature, in this paper we leverage recently proposed metrics of emergence to identify representations that display emergent properties. Specifically, we propose an end-to-end differentiable architecture that can learn maximally emergent representations of multivariate time series data. Our results show that causal emergence speeds up learning of more complex features of the data relative to pure mutual information maximisation, and we demonstrate the scalability of our method through an analysis of real-world brain activity data.

## 2 Methods

### 2.1 Quantifying emergence

Consider a scientist measuring a system of interest composed of $n$ parts, and let $X_t^i$ denote the state of part $i$ at time $t$. The information that the joint process carries together from $t$ to $t'$ can be quantified by the standard Shannon mutual information $I(\boldsymbol{X}_t; \boldsymbol{X}_{t'})$, where $\boldsymbol{X}_t = (X_t^1, ..., X_t^n)$ is the joint state of the system at time $t$.

How can one characterise an emergent macroscopic variable of such system? Following Ref. [43], one can define emergent variables $V_t$ as satisfying two key criteria:

(i) **Supervenience**: there exists a function (or *coarse-graining*) $f$ such that $V_t = f(\boldsymbol{X}_t)$.

(ii) **Unique information**: $V_t$ holds unique predictive information about the future evolution of the system $\boldsymbol{X}_{t'}$ that cannot be found in the individual parts $X_t^1, \ldots, X_t^n$ by themselves.

Critically, the unique information that $V_t$ holds about $\boldsymbol{X}_{t'}$ can be rigorously quantified using the framework of *Partial Information Decomposition* (PID, [54]), and its recent extension to time series data ($\Phi$ID, [33]). Emergence, therefore, is defined as the capability of a supervenient variable to provide predictive power that cannot be reduced to underlying microscale phenomena.

Quantifying unique information in high-dimensional systems can be highly non-trivial. Luckily, the $\Phi$ID formalism allows us to derive simpler measures that provide sufficient criteria for emergence. In particular, it has been shown that the following is a sufficient condition for causal emergence [43]:

$$\Psi := I(V_t; V_{t+1}) - \sum_i I(X_t^i; V_{t+1}) > 0 \, . \tag{1}$$

Importantly, $\Psi$ is comparatively easy to calculate, as it relies only on pairwise marginal distributions and on standard Shannon mutual information. These key features allow the framework to be applicable on a wide range of scenarios, as illustrated by the applications reviewed in Ref. [34]. Note that here we take $t' = t + 1$, but in principle any $t' > t$ is valid.

The reason why $\Psi > 0$ is only a sufficient, but not necessary, condition for emergence is that in some systems multiple $X_t^i$ can have the same information about $V_{t+1}$, and hence the sum of terms $I(X_t^i; V_{t+1})$ may 'double-count' information — resulting in a negative bias in $\Psi$. This double-counting can be alleviated by discounting from Eq. (1) the redundant information between the $X_t^i$'s about $V_{t+1}$. Here we do this using a measure of redundancy known as 'Minimum Mutual Information' [4], which yields the following 'adjusted' emergence criterion (Supp. Sec. C):

$$\Psi_A := I(V_t; V_{t+1}) - \sum_i I(X_t^i; V_{t+1}) + (n-1) \min_i I(X_t^i; V_{t+1}) > 0 \, . \tag{2}$$

### 2.2 Model architecture and information estimators

**Maximising emergence.** Our aim is to establish an automated procedure to identify emergent macroscopic variables $V_t$ with respect to a microscopic substrate $\boldsymbol{X}_t$. For this, we investigate parametric coarse-grainings $V_t = f_\theta(\boldsymbol{X}_t)$ that can be optimised to maximise $\Psi$ via a differentiable objective function.

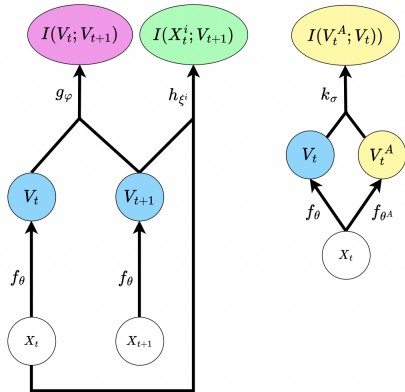

Figure 1: **Architecture for calculating loss terms for learning causally emergent representations**. A representation network $f_\theta$ applied to the data $\boldsymbol{X}_t$ learns a feature $V_t$. This feature is trained to optimise $\Psi$ made up of predictive and marginal mutual information terms estimated by $g_\varphi$ and $h_{\xi^i}$ respectively (left). A further critic $k_\sigma$ may be added to calculate the mutual information with another emergent feature $V_t^A$, to encourage the learning of a diverse set of emergent features (right).

A key ingredient to maximising $\Psi$ is employing a suitable estimator of Shannon's mutual information. Although many estimators of mutual information exist [27, 29], most are not differentiable, and thus not suitable for optimisation with standard representation learning architectures. Fortunately, a number of differentiable estimators of mutual information exist [39, 50].

We use the Smoothed Mutual Information "Lower-bound" Estimator (SMILE, [50]), which is one of a family of approaches that formulates mutual information estimation as a variational problem, and was specifically designed to address the issue of high variance in existing estimators such as NWJ [37] and MINE [5]. The SMILE mutual information estimator is given by

$$I(X;Y) = \mathbb{E}_{p(x,y)} \left[ \log \frac{p(x,y)}{p(x)p(y)} \right] \tag{3}$$

$$\geq \mathbb{E}_{p(x,y)} \left[ g_\varphi(x,y) \right] - \log \mathbb{E}_{p(x)p(y)} \left[ \mathrm{clip} \left( e^{g_\varphi(x,y)}, e^{-\tau}, e^{\tau} \right) \right] \triangleq I_\varphi^{\mathrm{S}}(X;Y), \tag{4}$$

where $g_\varphi$ is a parameterised function that estimates the log density ratio $\log \left( p(x,y)/(p(x)p(y)) \right)$, $\mathrm{clip}(v,l,u) = \max \left( \min \left( v, u \right), l \right)$ and $\tau \geq 0$ is a hyperparameter. As $\tau \to \infty$, $I^{\mathrm{S}}$ converges to the MINE estimation [5], but a finite $\tau$ prevents the potentially exponential growth of the variance of the estimate with MI, which MINE suffers from [50].

Equipped with this estimator, we can now formulate our representation learning algorithm for causally emergent features. The architecture is schematically shown in Fig. 1. Our main method involves three learnable functions:

1. A representation network $f_\theta$, that learns a supervenient variable $V_t = f_\theta(\boldsymbol{X}_t)$.

2. A critic for the macroscopic variable $g_\varphi$, that controls the estimation of $I(V_t; V_{t+1})$.

3. A set of critics for the microscopic variable $h_{\xi^i}$, that control the estimation of $I(X_t^i; V_{t+1})$. The number of critics equals the number of constituent atoms of the system.[1]

We will demonstrate that the representation critic can learn emergent features on an algorithmic dataset by maximising the following metric:

$$\Psi^{\mathrm{S}}(\theta, \varphi, \{\xi^i\}) := I_\varphi^{\mathrm{S}}(f_\theta(\boldsymbol{X}_t); f_\theta(\boldsymbol{X}_{t+1})) - \sum_i I_{\xi^i}^{\mathrm{S}}(X_t^i; f_\theta(\boldsymbol{X}_{t+1})) . \tag{5}$$

---

[1]Alternatively, one could consider using a single critic $h_\xi$ shared across all $X_t^i$. We found this option to be computationally cheaper but less effective.

We refer to $\Psi^S$ as the *emergence objective function*, and the first term in the RHS of Eq. (5) as the predictive mutual information [6] – since (by the data processing inequality [15]) it represents a lower bound on the joint mutual information between the past and future states of the whole system, $I(\boldsymbol{X}_t; \boldsymbol{X}_{t'})$. As a control condition, we also ran experiments with an objective function consisting of only the predictive information, removing the marginal mutual information terms. After training has converged, we ran 'post-hoc' tests to further verify that the learned feature is emergent by freezing the representation network $f_\theta$, retraining the critics to accurately estimate $I(V_t; V_{t+1})$ and $I(X_t^i; V_{t+1})$, and calculating $\Psi_A$. If this resulting $\Psi_A$ is positive, we conclude the feature is emergent. For convenience, pseudocode for the algorithm used to train the model is provided in Supp. Sec. A.

**Learning diverse emergent features.** A well-known fact about complex systems is that they can have more than one emergent property, and/or be described by multiple order parameters [25].

To learn a diverse set of emergent features, we consider a scenario where an emergent feature $V_t^A$ has been learned and its corresponding parameters $\theta^A$ are fixed. Intuitively, our goal is to find a new feature that is substantially different from the existing one by incentivising it to be statistically independent from $V_t^A$. To implement this, we add a penalty term $I_\sigma^S(f_{\theta^A}(\boldsymbol{X}_t); f_\theta(\boldsymbol{X}_t))$ to the objective function and include a fourth learnable function to our method: a critic $k_\sigma$ that estimates this mutual information, also using SMILE.

**Improving training stability.** An important point to note of the proposed method is that there is an adversarial relationship between the microscopic critics $h_{\xi^i}$ and the macroscopic critic $g_\varphi$ through the representation network $f_\theta$, reminiscent of the adversarial learning dynamics in GANs [16]. In essence, the former pushes the representation network $f_\theta$ to maximize information about the system's state, while the latter pushes it to minimize it. This could lead to potentially unstable learning dynamics and various failure modes – for example, a representation network could "trick" the microscopic critics into reducing their MI estimate by exploiting their mistakes, rather than genuinely reducing $I(X_t^i; V_{t+1})$, resulting in an artificially inflated value of $\Psi$.

Fortunately, unlike in GANs, it is not harmful for the critics to be overpowered compared to the representation learner[2] because the SMILE estimation is bounded from above by the true MI [50]. Following this reasoning, we *i*) update the critics multiple times for each representation learner update, using a higher learning rate and more parameters; and *ii*) pre-train the critics before we start training the representation learner, so that they provide a more robust MI estimation and better training signal. This results in slower, but more stable, learning dynamics for $f_\theta$.

Another method to increase training stability is to use a dedicated choice of the architecture of the representation network. $f_\theta$ first linearly projects the input to a higher-dimensional space, then passes it through an MLP with residual ('skip') connections, and then projects it linearly down to the output vector $V_t$. Because of the skip connections (and because of the auto-correlation of $\boldsymbol{X}_t$), at initialisation the output already contains useful information about the system and yields a high value of the macroscopic information $I_\varphi^S(f_\theta(\boldsymbol{X}_t); f_\theta(\boldsymbol{X}_{t+1}))$ before $\theta$ has been trained. Therefore, this allows us to reduce the contribution of this term in the objective function, and thus reduce the influence of $g_\varphi$ on $f_\theta$.[3]

## 2.3 Datasets

### 2.3.1 Synthetic datasets

**Bit-string dataset.** We evaluate our method for learning causally emergent representations by applying it to sequences of random bit-strings of length $n$ with two constructed temporal correlations:

1. The *parity* of the first $n-1$ bits is auto-correlated across time, such that

$$\mathbb{P}\{ \oplus_{i=1}^{n-1} X_{t+1}^i = \oplus_{i=1}^{n-1} X_t^i \} = \gamma_{\text{parity}} > \frac{1}{2} ,$$

where $\oplus$ represents modulo-2 addition.

---

[2]In GANs this could lead to mode collapse [51].

[3]Although this could in principle lead to learned representations with reduced predictive information, in practice we observe this leads to increased stability and only small decrements in $I(V_t; V_{t+1})$.

2. The last (or *extra*) bit in the bit-string $X^n$ is auto-correlated across time, such that

$$\mathbb{P}\{X_{t+1}^n = X_t^n\} = \gamma_{\text{extra}} > \frac{1}{2} .$$

Since parity is a synergistic function of the bits of a bit-string (i.e. it cannot be predicted from any of the input bits individually [42]), and since the parity predicts some information about the future evolution of the system, $V_t = \oplus_{i=1}^{n-1} X_t$ is an emergent feature of the system.

Despite its simplicity, this dataset has two important advantages: there is a known emergent feature (the parity), and one can calculate the mutual information and the emergence measure analytically.[4] These properties will allow us to verify that the model has successfully extracted the expected emergent properties and that mutual information is being accurately estimated.

**Conway's Game of Life.**    To further evaluate our method on a more complex system, we apply it to simulations from Conway's Game of Life (GOL) – a canonical setting for the study of emergence [1]. GOL is a cellular automaton on an $N \times N$ grid where each cell's state evolves based on simple, deterministic rules. We initialize a "glider" pattern [19] at a random position within the grid, and this glider moves diagonally by cycling through four distinct states until it reaches a boundary and the simulation concludes. The dataset consists of multiple such simulations concatenated to form a continuous sequence. To effectively capture the spatial dependencies inherent in the Game of Life, we replace the multilayer perceptron used in previous experiments with a convolutional neural network for the representation learner $f_\theta$, allowing us to exploit the 2D structure of the grid-based simulations.

### 2.3.2   Real world brain datasets

**Primate ECoG dataset.**    We evaluate our method on a dataset of electrocorticography (ECoG) brain activity data from a macaque monkey, originally reported by Chao *et al.* [10]. The dataset contains long time series of electrical activity measured with 64 electrodes placed across the monkey's brain surface. We minimally pre-process the data by applying a second-order Butterworth high-pass filter with a 1 Hz cutoff, downsampling the signals to 300 Hz, and finally standardising the data before applying our method to learn emergent features from it.

**Human MEG dataset.**    We evaluate our method on a magnetoencephalography (MEG) brain activity dataset from healthy human participants, originally reported in four pharmaco-MEG studies [36]. The dataset includes resting-state recordings collected before the administration of various pharmacological agents. MEG signals were sampled at 600 Hz with a 0–300 Hz bandpass filter. For our analysis, we focus on the placebo condition (pre-drug administration) and train our model $f_\theta$ on data from multiple subjects, using each epoch to represent a different participant to capture features common across individuals. Data was pre-processed following standard procedure [9] and standardised before applying our method to learn emergent features.

**Human fMRI dataset.**    We also evaluate our method on a functional magnetic resonance imaging (fMRI) dataset from 100 unrelated human participants from the "minimally preprocessed" release by the Human Connectome Project [53, 20]. We performed additional preprocessing following Luppi *et al.* [31] and calculated 100 time series capturing the activity of each of the regions in the Schaefer brain atlas [45]. We further standardise the data before applying our method to learn emergent features from it.

## 3   Results

In this section, we discuss results on learning emergent features on the synthetic and real world datasets introduced above, as well as comparisons to baselines and some ablation studies. All experiments can be run on a single A10G GPU in less than two hours.

---

[4]Specifically, the emergence metric is $\Psi = 1 - H_2(\gamma_{\text{parity}})$ and the predictive information is $I(X_t; X_{t+1}) = 2 - H_2(\gamma_{\text{parity}}) - H_2(\gamma_{\text{extra}})$, where $H_2(p)$ represents the entropy of a Bernoulli distribution with parameter $p$. Here we set $\gamma_{\text{parity}} = \gamma_{\text{extra}} = 0.99$, and show results for other $\gamma_{\text{parity}}, \gamma_{\text{extra}}$ values in the Appendix.

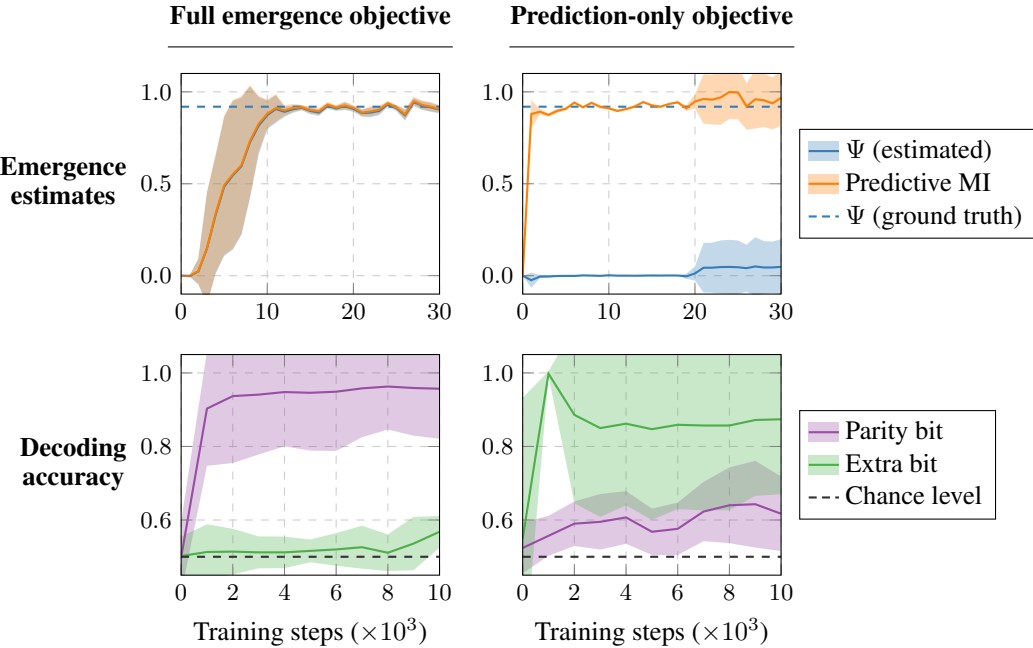

Figure 2: **The proposed architecture recovers ground truth emergent features**. Using the emergence objective function (left column), the model finds the correct $\Psi$ value and is able to recover the known emergent feature (parity bit). Using only predictive MI as the objective (right column), the model fails to discover any emergent features.

## 3.1 Learning emergent features in synthetic datasets

**Bit-string dataset.** Results show that our proposed architecture can accurately estimate the ground-truth value of $\Psi$ in the synthetic dataset, confirming it is able to learn causally emergent representations (Fig. 2). To interpret the contents of the learned representation, we trained decoders with standard supervised learning to predict both the parity of the first $n - 1$ bits (the parity bit) and the last auto-correlated bit (the extra bit). We found that the parity bit could be decoded with high accuracy but the extra bit could not, confirming that the learned representation indeed corresponded to an emergent feature. For additional verification, we fixed the representation learner's weights $\theta$ and re-trained both critics to estimate both $\Psi$ and $\Psi_A$, which were both positive and confirmed our results.

As expected, when the marginal MI terms are removed from the objective function (Fig. 2, right column), the model is no longer able to obtain the correct $\Psi$ value – and, interestingly, only the extra bit (but not the parity bit) is encoded in the representation. We hypothesise that, in the absence of the regularisation induced by the marginal MI, the system's inductive biases lead it towards learning "low-order" (i.e. non-emergent) representations. Note that, despite having a constraint removed, the model without marginal MI loss is unable to extract the full predictive information of the system (which equals approximately $1.84\,\mathrm{bit}$), showing that using the full emergence loss could incentivise the system to learn features that provide information about the system's dynamics that would otherwise be ignored (we elaborate on this further in Section 3.3). We obtain qualitatively similar results with a noisier version of the same data generating process (Supp. Fig. 5).

**Conway's Game of Life.** By employing a convolutional neural network for the representation learner $f_\theta$, we successfully learned emergent features from this dataset, evidenced by a positive $\Psi_A$ value after freezing the representation network and retraining the critics (Supp. Fig. 8a). To interpret the emergent feature, we attempted to predict the glider's position and state from the representation. While the position prediction performed at chance level, the state prediction achieved an accuracy of $53\%$, significantly higher than the random baseline of $25\%$ (Supp. Fig. 8b). This suggests that the emergent feature encodes information about the glider's state. When $f_\theta$ is trained with a prediction-only objective, the state prediction achieved an accuracy of $35\%$, suggesting the learned feature encodes less information about the state of the glider when trained on this objective.

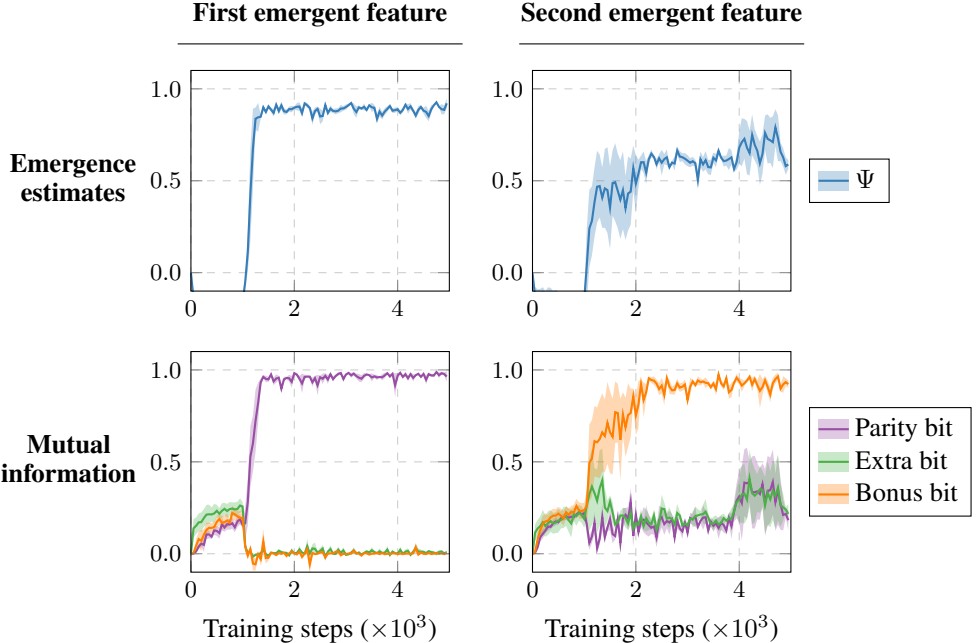

Figure 3: **Our method learns a diverse set of multiple emergent features from the same system**. Training two representation learners to learn independent emergent feature on the synthetic dataset. Both learned features were emergent (top row). The first learner (left column) yielded a feature that has high mutual information with the system's parity bit (bottom left), while for the second learner (right column) it had high mutual information with the bonus bit (bottom right).

These findings demonstrate that our approach can learn meaningful emergent features in complex, high-dimensional systems like the Game of Life, capturing collective behaviors that are not localized to individual components.

### 3.2 Learning diverse features in the bit-string dataset

As our next step, we set out to test our algorithm to learn diverse features in the same bit-string dataset. For this set of experiments we used the representation learner with skip connections described in Section 2.2, ran our algorithm as above until convergence, and fixed the representation learner's weights (denoted by $\theta^A$). We then trained a second representation learner with the regular $\Psi^S$ objective plus a penalty term $I(V_t; V_t^A)$ to obtain a new feature that is statistically independent from the parity bit.

Interestingly, this process revealed a new, unexpected feature that was not originally designed (Fig. 3): the XOR of the parity and extra bit (which we will refer to simply as the *bonus* bit). This bit is emergent (since it cannot be predicted by any of the $X_t^i$ and is auto-correlated, since both the parity and extra bits are), albeit with a lower $\Psi$ than the parity bit – which fits well with the result that our model did not learn this feature spontaneously. This shows the capability of our method to discover new aspects of a system under study, even in the case of simple and explicitly constructed systems.

### 3.3 Comparison to baselines and ablation studies

**Comparison with RNNs.** In order to investigate whether the failure of the prediction-only baseline to learn emergent features was to do with the capacity of the architecture, we trained a standard RNN on the bit-string dataset using mean squared error (MSE) loss, with its hidden state serving as the representation for decoding. The hidden state dimension matched that of our emergent feature network $f_\theta$ for a fair comparison. As shown in Fig. 4, the RNN consistently learned the non-emergent extra bit but encoded negligible information about the two emergent bits, resulting in $\Psi_A = 0$. This indicates that the RNN did not capture any emergent features.

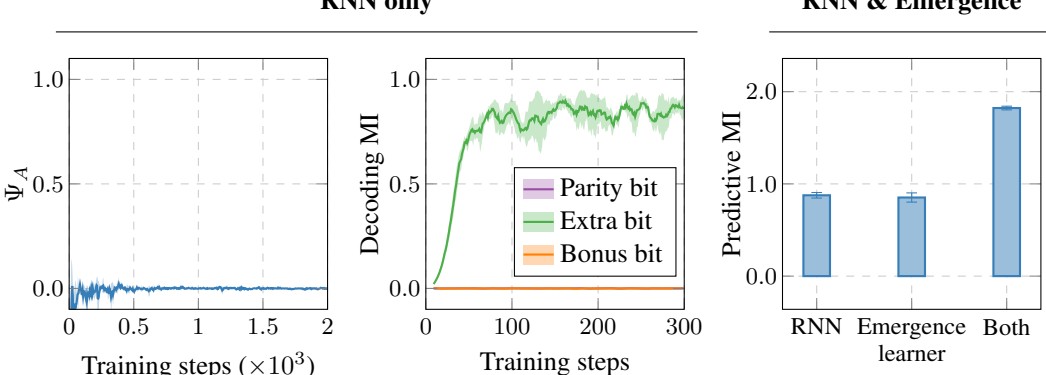

Figure 4: **Standard methods do not learn emergent features, and their performance increases when combined with emergent features.** The hidden state of an RNN trained on the bit-string dataset has negligible $\Psi_A$, indicating no emergent feature learned (left). Accordingly, the mutual information between the hidden state and the extra, parity, and bonus bits shows that only the non-emergent extra bit is encoded (middle). Interestingly, representations learned by an RNN and by our method can be combined to yield better predictions of the future state of the system (right).

**Combining emergent and non-emergent features.** The fact that standard architectures learn representations that are predictive, yet not emergent, suggests they may be learning different aspects of the data compared to our method. To test this hypothesis, we investigated whether combining emergent features from our method with representations from standard models would enhance the predictive performance of either. As a proof of concept, we trained both an RNN and our emergence learner $f_\theta$ on the bit-string dataset, and compared the mutual information between $X_{t'}$ and:

1. The hidden state of the RNN.
2. The emergent feature $V_t$ learned by $f_\theta$.
3. The concatenation of $V_t$ with the hidden state of an RNN.[5]

As shown in Fig. 4, the combined representation encoded more information about the future state of the system, $X_{t'}$, than either the RNN or the emergence learner alone. While this result is only shown on a very simple synthetic dataset, it suggests that our emergence learner extracts meaningful features that are different from, and can be effectively combined with, other representation learning architectures. We hypothesise that standard representation learning techniques can more effectively capture microscale properties, and since our method captures macroscale properties, both can be naturally combined to enhance performance in downstream tasks.

**Ablations to emergence loss.** The main emergence loss function we propose in this work has three terms: a predictive MI term estimating $I(V_t; V_{t'})$, marginal MI terms estimating $I(X_t^i; V_{t'})$, and a diversity term estimating $I(V_t^A; V_t)$. In order to understand the contribution of each term to the behaviour of the model, we performed ablation studies removing each one of these. We describe each ablation in turn:

1. As seen in Sec. 3.2 (and later in Sec. 3.4), ablating the $I(V_t; V_{t'})$ term from $\Psi$ objective results in the network learning features that are not as correlated in time as they would otherwise be, but that still satisfy the criteria for emergence. Therefore, our proposed architecture with skip connections may learn slightly less emergent features, but with the advantage of more stable training dynamics.

2. Ablating the $I(X_t^i; V_{t'})$ terms objective results in an algorithm that only maximises predictive information $I(V_t; V_{t'})$, akin to the Deep InfoMax method [23]. As seen in Sec. 3.1, and as expected, training without the marginal terms results in representations that are not emergent.

[5]We retrained both models using half the representation size, to ensure a fair comparison between all three cases.

3. Ablating the diversity term in the experiments in Sec. 3.2 and training multiple runs on the bit-string dataset results in the representation network learning the extra bit $91\%$ of runs, and the bonus bit $9\%$ of runs. This implies there are strong inductive biases towards which emergent features are learned, and it is thanks to the diversity term that we can reliably recover both.

## 3.4 Learning emergent features in brain activity data

Finally, we demonstrate the scalability of our method by learning emergent features in three types of high-dimensional brain activity datasets – ECoG, MEG, and fMRI – to evaluate its ability to learn emergent features across different neural recording modalities and spatial scales.

**ECoG data.** For the ECoG dataset [10], it was crucial to use skip connections in the representation learner, since the method as used in Section 3.1 was consistently unable to find emergent features (despite their known presence in this dataset [43]).

With this configuration, our method is able to successfully learn emergent representations of the ECoG data (Supp. Fig. 9a). The resulting feature was verified to be emergent by a positive post-hoc $\Psi_A$ test. Interestingly, using an $f_\theta$ with skip connections we obtained emergent features even when the macroscopic MI term $I(V_t; V_{t+1})$ was completely removed from the objective function, which further simplified learning dynamics. Empirically, we found that this does not cause $I_\varphi^S(f_\theta(\boldsymbol{X}_t); f_\theta(\boldsymbol{X}_{t+1}))$ to decrease over training substantially, suggesting that using just the marginal MI terms is a good objective function to remove information about the parts while preserving information about the whole that was present at initialisation.

As a final experiment, we also found that our method could learn a second emergent feature, also verified with a post-hoc $\Psi_A$ check (Supp. Fig. 10).

**MEG data.** We extended our analysis to magnetoencephalography (MEG) recordings from healthy human participants [36]. Training on data from multiple subjects, our method readily learned emergent features, as evidenced by a positive post-hoc $\Psi_A$ check (Supp. Fig. 9b). The consistency of emergent features across different individuals suggests that our model captures fundamental aspects of neural dynamics common to human brain activity.

Notably, emergent features were learned more easily from MEG data compared to ECoG data, potentially due to differences in spatial resolution and the nature of the recorded signals.

**fMRI data.** Finally, we applied our method to functional magnetic resonance imaging (fMRI) data from 100 unrelated human participants [45]. Despite the coarser spatial and temporal resolution of fMRI, our method was able to learn emergent features, as indicated by a positive $\Psi_A$ value (Supp. Fig. 9c). This demonstrates that emergent dynamics are present even at the macroscopic scale of brain activity. On this dataset, we also determined that a standard MLP did not learn emergent features in its representation (see Supp. Sec. B.2 for more details).

The successful extraction of emergent features across all three datasets highlights the robustness of our method. It underscores its capability to uncover collective neural dynamics in complex, high-dimensional brain activity data, regardless of the recording modality or spatial scale.

Table 1: **Our method learns emergent features in multiscale datasets of brain activity**. Final values of $\Psi$ and $\Psi_A$ as found during the training run of $f_\theta$ and post-hoc evaluation respectively. All values are greater than zero, indicating that in each case an emergent feature has been found.

|  | ECoG | MEG | fMRI |
|---|---|---|---|
| Training $\Psi$ | $0.8 \pm 0.12$ | $3.0 \pm 0.30$ | $2.7 \pm 1.3$ |
| Post-hoc $\Psi_A$ | $1.2 \pm 0.19$ | $2.6 \pm 0.52$ | $3.3 \pm 0.44$ |

# 4 Related work and future directions

The method we have proposed here is part of a growing literature leveraging methods from information theory to enhance deep learning architectures, and representation learning algorithms in particular. A small, far from exhaustive list of some noteworthy examples includes InfoNCE [38], deep variational information bottleneck [2], $\beta$-VAE [8], or TC-VAE [11]. In future work, it would be interesting to see if our method is competitive or can be combined with recent representation learning methods on standard benchmarks [22, 38, 52].

In practice, one of the challenging aspects of this method is to overcome the instability of training to make sure a valid emergent solution is found (i.e. one where the post-hoc check still shows $\Psi_A > 0$). One potential approach is to incorporate stabilisation techniques from the GAN literature (such as e.g. spectral normalisation [35]). Another potential approach is to use a mutual information upper bound (instead of our current lower bound) for the marginal MI terms. Although mutual information upper bounds are not as common as lower bounds, there are a few options available [13, 12]. In practice, we found both of these to be less effective than our skip connection method, but they remain a promising avenue for future work.

There is a growing body of machine learning research that engages directly with PID, the information-theoretic backbone of our emergence theory. For example, recent work has proposed a differentiable redundancy measure [32], which has been used as an objective function to train deep neural networks [21]. Alternatively, there are also methods that estimate redundancy using deep neural networks [26], analogous to SMILE. Although none of these estimators can be directly applied to estimate the unique information that underlies the definition of emergence (Section 2.1), we expect that extensions of these exciting developments will also open new possibilities in the study of emergence.

Finally, it is worth mentioning that there are a number of other approaches that focus on different aspects of emergence (see Supp. Sec. D). The approach presented here was chosen for two reasons: (i) its intuitive nature, focusing on how emergent properties arise from collective interactions that cannot be fully explained by examining components in isolation; and (ii) the existence of efficient, scalable, and differentiable proxies for its estimation. Constructing similar methods as the one proposed here for other metrics of emergence would be a challenging but extremely interesting line of future work.

# 5 Conclusion

Emergence, the phenomenon whereby a system becomes 'more than the sum of its parts', is an promising conceptual tool to investigate cognitive processes in artificial and biological systems. In this paper, we proposed a machine learning method for discovering emergent variables in time series data that leverages a recent information-theoretic characterisation of emergence [43] and advances in mutual information estimation from data with neural networks [50].

We first showed in a synthetic dataset that our method can estimate emergence and successfully discover a known emergent feature. Interestingly, a pure information maximisation objective struggled to learn this feature, suggesting that our method facilitates the identification of complex features of the data. Furthermore, we also proposed a slight modification of our method that can learn a diverse set of features from the same system. Finally, we also showed that our method can scale up to learn emergent features in real-world brain activity data.

Overall, our method opens up a range of possibilities for the practical study of emergence, as well as for other machine learning problems more broadly. For example, our method may be explored in conjunction with other representation learning algorithms to capture aspects of complex systems that are otherwise difficult to learn. From an application perspective, we hope this method can be further leveraged in neuroscience to reveal new aspects of brain function.

**Software availability**

Code implementing our proposed architecture and reproducing our key results is available at https://github.com/Imperial-MIND-lab/causally-emergent-representations

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

# A  Model pseudocode

**Algorithm 1:** Pseudocode for the causally emergent representation learning algorithm.

**Input:** Dataset $\mathcal{D} = \{(x_t, x_{t+1})\}$, *num_epochs*, number of atoms $n$, Optional[pretrained representation network $f_{\theta^A}$]

Initialize representation network $f_\theta$, predictive critic $g_\phi$, marginal critics $\{h_{\xi^i}\}^n$, Optional[diversity critic $k_\sigma$]

**for** *epoch* $\leftarrow 1$ **to** *num_epochs* **do**
    **for** $x_t, x_{t+1}$ *in* $\mathcal{D}$ **do**
        $v_t, v_{t+1} \leftarrow f_\theta(x_t), f_\theta(x_{t+1})$
        $\hat{I}_{v_t, v_{t+1}} \leftarrow g_\varphi(v_t, v_{t+1})$
        Update $g_\varphi$ to maximise SMILE lower bound (Eq. 4)
        **for** $i \leftarrow 1$ **to** $n$ **do**
            $\hat{I}_{x_t^i, v_{t+1}} \leftarrow h_{\xi^i}(x_t^i, v_{t+1})$
        **end**
        Update $\{h_{\xi^i}\}^n$ to maximise SMILE lower bound (Eq. 4)
        $\hat{\Psi} = \hat{I}_{v_t, v_{t+1}} - \sum_{i=1}^n \hat{I}_{x_t^i, v_{t+1}}$
        **if** *training diverse emergent feature* **then**
            $\hat{I}_{v_t, f_{\theta^A}} \leftarrow k_\sigma(v_t, f_{\theta^A}(x_t))$
            loss $\leftarrow -\hat{\Psi} + \hat{I}_{v_t, f_{\theta^A}}$
        **else**
            loss $\leftarrow -\hat{\Psi}$
        **end**
        Update $f_\theta$ to minimize loss
    **end**
**end**

# B  Supplementary results

## B.1  Supplementary figures

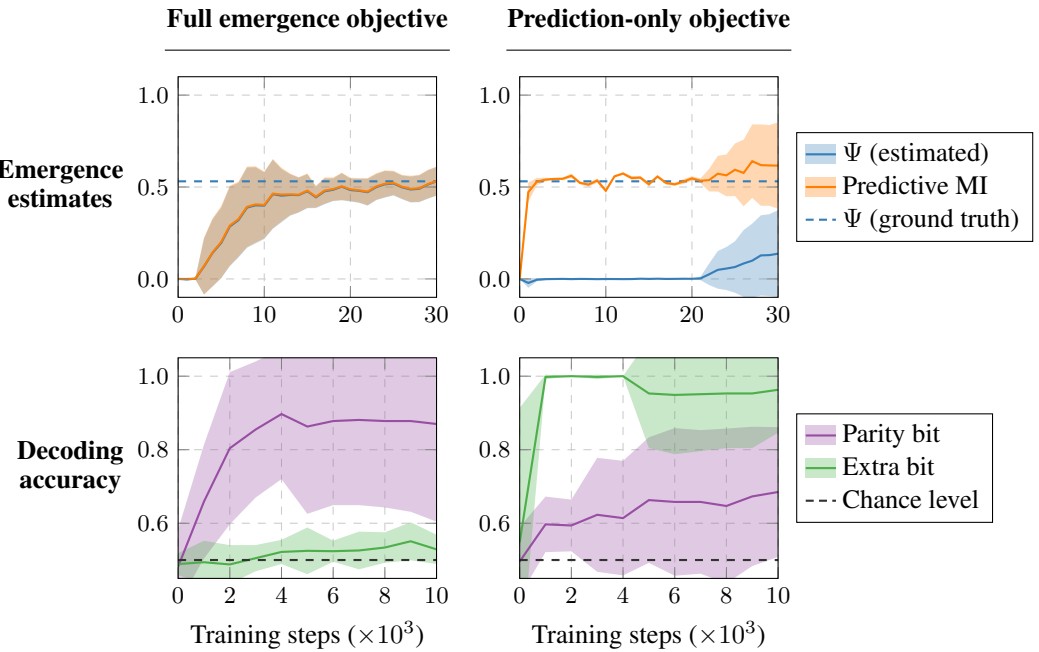

Figure 5: **Replication of main results with noisier data**. Same as in Fig. 2, but with $\gamma_{\text{parity}} = \gamma_{\text{extra}} = 0.9$.

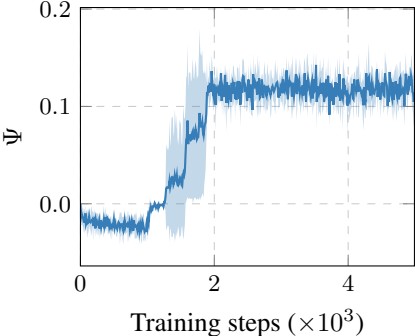

Figure 6: **Replication of main results with even noisier data**. Emergence values on the bit-string dataset with $\gamma_{\text{parity}} = \gamma_{\text{extra}} = 0.7$.

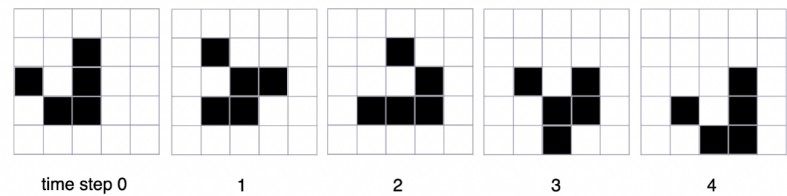

Figure 7: **Glider states in the Game of Life simulation**. Gliders cycle deterministically between these four states. To interpret the learned emergent feature in the experiments in Section 3.1, we trained a standard classification MLP to predict these four states from the value of $V_t$.

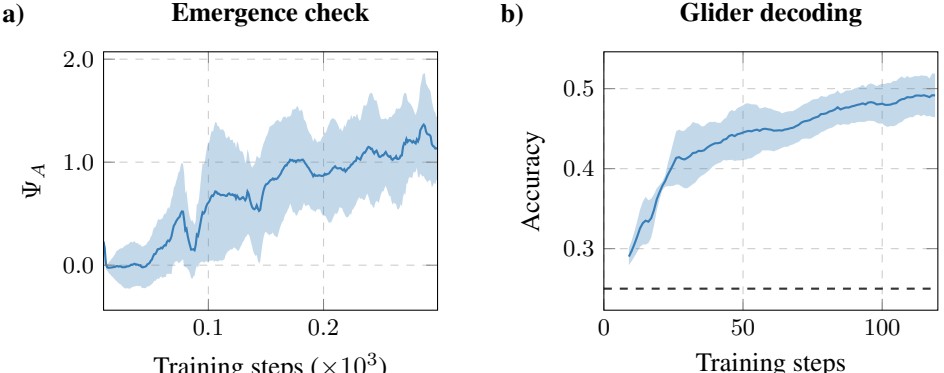

Figure 8: **Our method learns interpretable emergent features in Conway's Game of Life**. **a)** Post-hoc checks reveal $\Psi_A > 0$, confirming the learned feature is emergent. **b)** Classification accuracy of the state of the glider (c.f. Supp. Fig. 7) on a held-out test set. Dashed black line represents chance level at 25%.

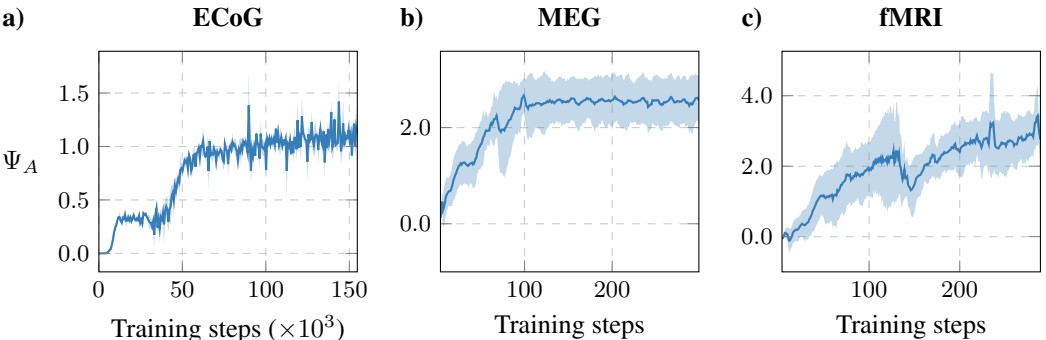

Figure 9: **Learning emergent features from brain activity datasets**. Post-hoc $\Psi_A$ checks for our representation learner trained on **a)** primate ECoG data, **b)** human MEG data, and **c)** human fMRI data. Emergent features were successfully learned in all cases.

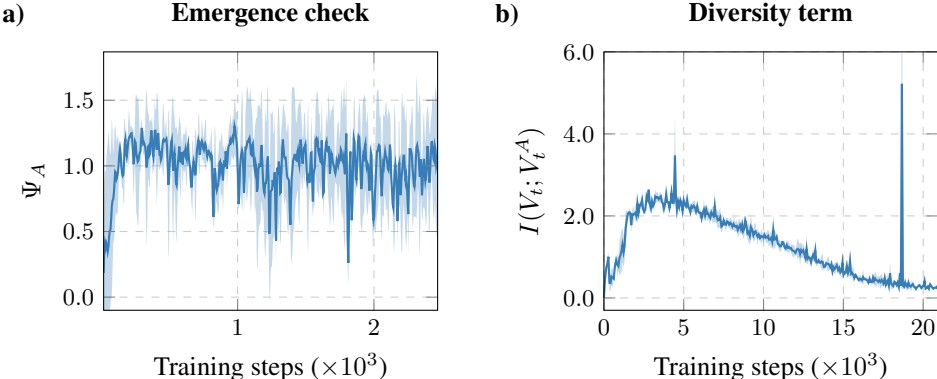

Figure 10: **Learning diverse emergent features from ECoG brain activity data**. **a)** After learning one emergent feature on the ECoG data (c.f. Supp. Fig. 9, adding the diversity loss term and training a new representation network results in another emergent feature, verified by a post-hoc $\Psi_A$ check. **b)** We further verify that the new feature $V_t$ is nearly independent from the previous one $V_t^A$, as their mutual information (estimated with SMILE and the critic $k_\sigma$) vanishes.

## B.2 MLP comparison on fMRI Dataset

In order to investigate further whether other representation learning methods could naturally learn emergent features, we also trained an encoder-decoder pair of multilayer perceptrons (MLPs) on the fMRI dataset to predict $X_{t'}$ from $X_t$ using an MSE objective. The encoder mapped $X_t$ to a representation $V_t$ with the same dimension as $f_\theta$'s output. A post-hoc test revealed that the encoder did not learn an emergent feature, as $\Psi_A < 0$ (after training, the model stabilised at $\Psi_A = -28 \pm 5$), indicating that standard autoencoder-like architectures may not capture emergent properties in complex data.

## C  Adjusted emergence metric

Let us present an informal derivation of the correction that leads to the refined criteria for emergnece presented in Eq. 2. This formulation relies on the Partial Information Decomposition framework [54], on which our emergence metric $\Psi$ is based.

Let us simplify the notation and use $B = V_{t+1}$ for the target variable and $A_i = X_t^i$ for the $i$-th source of information. Taking inspiration in Ref. [43, Appendix A], the total information about $B$ provided by $\boldsymbol{A} := (A_1, \dots, A_n)$ can be expressed as

$$I(\boldsymbol{A}; B) = \texttt{Red}(\boldsymbol{A}; B) + \texttt{Un}(\boldsymbol{A}; B) + \texttt{Syn}(\boldsymbol{A}; B) , \tag{6}$$

where $\texttt{Red}(\boldsymbol{A}; B)$ is the redundancy between sources, $\texttt{Syn}(\boldsymbol{A}; B)$ is their synergy, and $\texttt{Un}(\boldsymbol{A}; B)$ accounts for the unique information provided by each predictor and not the others. Of these, the synergistic component is the one associated to emergence by the theory in Ref. [43]. By using this decomposition and the chain rule of the mutual information [15], one can find that $I(A_i; B) = \texttt{Red}(\boldsymbol{A}; B) + \Theta_i$ where $\Theta_i$ accounts for other terms unique to source $i$, and hence

$$\sum_{i=1}^{n} I(A_i; B) = n \, \texttt{Red}(\boldsymbol{A}; B) + \sum_{i=1}^{n} \Theta_i . \tag{7}$$

This shows that, when summing the marginal mutual informations, the term $\texttt{Red}(\boldsymbol{A}; B)$ gets double-counted $n$ times, resulting in $\Psi$ containing this redundancy term $1 - n$ times. Hence, it is natural to propose a refined criteria for emergence that avoids this double-counting, given by

$$\Psi_A = I(V_t; B) - \left( \texttt{Red}(\boldsymbol{A}; B) + \sum_{i=1}^{n} \Theta_i \right) = \Psi + (n-1)\texttt{Red}(\boldsymbol{A}; B). \tag{8}$$

# D   Alternative Formalizations of Emergence

Several frameworks have been proposed to formalize the concept of emergence in complex systems. Below, we provide a brief overview of some approaches that are particularly well-suited to investigate time-series.

## D.1   Emergence as dynamical independence

Barnett [3] introduces a formalization of emergence based on information theory, defining an emergent macroscopic process $V_t = f(X_t)$ as one where the dynamics of $V_t$ are conditionally independent of the microscopic history $X_t^-$, given its own history $V_t^-$. Accordingly, the condition for emergence is defined using transfer entropy [46, 7] as follows:

$$T_t(X \rightarrow V) = I(V_t; X_t^- \mid V_t^-) = 0. \tag{9}$$

This framework captures the idea that the macroscopic state's future is determined by its own history, and additional knowledge of the microstate does not enhance prediction.

## D.2   Emergence via computational mechanics

Computational mechanics [17, 18, 48] defines emergence through the concept of *causal states*, which are equivalence classes of histories $\overleftarrow{S}$ leading to the same conditional distribution over futures $\overrightarrow{S}$. This leads to establish two keys quantities: the statistical complexity $C_\mu$ and excess entropy $\mathbf{E}$, which are calculated as

$$C_\mu = H(\mathcal{S}), \quad \mathbf{E} = I(\overleftarrow{S}; \overrightarrow{S}). \tag{10}$$

Above, the excess entropy quantifies how much information is shared between past and future, and the statistical complexity measures how much information processing is needed to enable such prediction.

Building on these ideas, emergence has been characterised in different ways. One approach has been to identify a process $S_t' = f(\overleftarrow{S}_t)$ as emergent when it increases the *efficiency of prediction* $e = \mathbf{E}/C_\mu$ compared to the microscopic description [48, Sec. 11.2]. Another approach has been to assign emergence when there is a new type of dynamics observed at the macroscopic level from the one observed at the micro [17, 44]; for example, dynamical patterns associated to a different computational class [17, Sec. 5.1]. Finally, another approach has been to ascribe emergence to macro-processes whose causal states are a coarse-graining of the causal states of the micro [41], which leads to a view closely related with the one presented in Section D.1.

## D.3   Granger emergence

Seth [47] proposes *Granger emergence*, where a macroscopic process $V_t$ is emergent with respect to the microscopic process $X_t$ if it satisfies the following conditions:

$$I(V_t; V_{t'} \mid X_t) > 0 \quad \text{and} \quad I(X_t; V_{t'} \mid V_t) > 0. \tag{11}$$

The first condition ensures that $V_t$ contains predictive information about its future that is not present in $X_t$, while the second condition establishes that $V_t$ arises from $X_t$.

## D.4   Emergence via effective information

Hoel et al. [24] introduce the concept of *effective information*, which measures the strength of dependencies using maximum entropy distributions [14]. More specifically, for a given probability kernel corresponding to $p(X_{t+1}|X_t)$, the corresponding effective information is $\text{Ef}(X) = I(X_t; X_{t+1})$ where $X_t$ is assumed to follow a uniform distribution. Then, emergence is assigned when the kernel $p(V_{t+1}|V_t)$ that results when applying the coarse-graining $V_t = f(X_t)$ satisfies

$$\text{Ef}(V) > \text{Ef}(X). \tag{12}$$

In other words, there is more information going through the macroscopic than the microscopic level when considering maximum-entropy inputs to both of them.

### D.5 Comparison with our framework

While these alternative formalisms provide valuable insights into emergence, they differ from the approach we use, which emphasizes the part-whole relationships by considering how the emergent feature encodes information about the microscopic components, particularly focusing on synergistic interactions. Our framework aligns with the view that emergent properties arise from collective interactions among parts that cannot be fully understood by examining components in isolation. Moreover, it is important to highlight that the metrics associated with most of these approaches are computationally expensive, and don't scale well with system size. Enabling similar methods as the one proposed here but with these other metrics of emergence, and comparing the resulting findings, would be an extremely interesting line for future work.

## E Hyperparameters

Hyperparameters for the models trained on the synthetic and brain datasets can be seen in the below tables. The hyperparameters do not change when learning diverse sets of emergent features. For each training run the clip was fixed at $5$ in the SMILE mutual information estimators and the optimizer used was AdamW.

Table 2: Hyperparameters for causal emergence representation learning on synthetic data

| Hyperparameter | Value |
|---|---|
| Number of bits in $X_t$ | 5 + 1 |
| Number of training steps | 5e6 |
| Parity autocorrelation $\gamma_{\text{parity}}$ | 0.99 |
| Extra bit autocorrelation $\gamma_{extra}$ | 0.99 |
| Batch size | 1000 |
| $f_\theta$ layer sizes | [256, 256, 1] |
| Microscopic variable critic layer sizes | [512, 512, 512, 256] |
| Macroscopic variable critic layer sizes | [512, 512, 512, 256] |
| Number of steps between $f_\theta$ updates | 5 |
| $f_\theta$ weight decay | 1e-3 |
| $f_\theta$ learning rate | 1e-4 |
| Critic learning rate | 1e-3 |

Table 3: Hyperparameters for causal emergence representation learning on ECoG data

| Hyperparameter | Value |
|---|---|
| Number of variables in $X_t$ | 64 |
| Number of training epochs | 60 |
| Batch size | 1000 |
| $f_\theta$ layer sizes | [256, 256, 256, 256, 256, 3] |
| Number of batches to pretrain critics | 300 |
| Microscopic variable critic layer sizes | [512, 512, 512, 256, 32] |
| Macroscopic variable critic layer sizes | [512, 512, 512, 32] |
| Number of steps between $f_\theta$ updates | 5 |
| $f_\theta$ weight decay | 1e-3 |
| $f_\theta$ learning rate | 1e-4 |
| Critic learning rate | 1e-3 |

Table 4: Hyperparameters used to train emergent feature network on MEG and fMRI dataset with $\Psi$

| Hyperparameter | Value |
| --- | --- |
| Batch size | 1000 |
| Macroscopic variable critic layer sizes | [512, 512, 512] |
| Microscopic variable critic layer sizes | [512, 1,024, 1,024, 512, 32] |
| Critic learning rate | 1e-3 |
| Number of training epochs | 5 |
| $f_\theta$ layer sizes | [256, 256, 256, 256, 256, 32, 3] |
| $f_\theta$ learning rate | 1e-4 |
| $f_\theta$ weight decay | 1e-6 |
| Number of steps to pretrain critics | 150 |
| Number of steps between $f_\theta$ updates | 5 |

Table 5: Hyperparameters for Game of Life emergent feature learning with $\Psi$ criterion

| Hyperparameter | Value |
| --- | --- |
| Grid size | $15 \times 15$ |
| Batch size | 1000 |
| Number of training epochs | 10 |
| Feature size | 1 |
| Number of steps to pretrain critics | 150 |
| Number of steps between $f_\theta$ updates | 5 |
| Convnet encoder layer sizes | Conv[32, 64, 128, 256], FC[256, 128, 3] |
| Downward critic learning rate | 1e-3 |
| Macroscopic variable critic layer sizes | [512, 512, 512, 32] |
| Microscopic variable critic layer sizes | [512, 512, 512, 32] |
| Critic learning rate | 1e-3 |

