# OpenReview forum: "Learning diverse causally emergent representations from time series data"
_NeurIPS.cc/2024/Conference — NeurIPS 2024 poster_

### Official Review · Reviewer_7mVS · 2024-07-07

**Soundness:** 3
**Presentation:** 2
**Contribution:** 3
**Rating:** 5
**Confidence:** 3

**Summary:**

The article proposes a learning scheme aimed at detecting emergent quantities from time series data of systems made of many interacting parts, such as the brain. To this end the authors combine "minimum mutual information", a previously introduced emergence criterion, with SMILE, a differentiable lower bound estimator for mutual information. Differentiability is crucial for the loss function to be optimizable efficiently. They apply this architecture to two examples: First, a series of random bit strings with time-correlated parity, where parity is considered the emergent quantity. Second, real-world data of macaque brain activity. The approach successfully identifies parity in the first example. The authors claim that an emergent feature has been learned for the second example also.

**Strengths:**

While the individual parts of the learning scheme are not new, their combination into a differentiable architecture is original and seems like a promising direction to me. The analysis seems sound, even though I found the presentation at times a bit hard to follow as some parts seem to be missing. The individual quantities are mostly clearly defined and the individual results are statistically significant in terms of error bars.

**Weaknesses:**

From the article alone, I could not fully understand the architecture and its training procedure that is illustrated in Fig. 1. I could not find code that would reproduce it, or a detailed pseudo-code description of the algorithm. It is unclear to me what emergent feature was found for the monkey example, or how Fig. 4 proves that any such feature was found. While the architecture and direction seems promising to me, a few more benchmarks would help make the case that this scheme can find emergent features in many settings. The two examples shown are one toy example with unnatural time dynamics and one real world example where it is hard to understand the dynamics from first principles. Benchmarking this new method on more standard examples with emergent behavior, such as Ising models, would be more convincing.

**Questions:**

1) How does Figure 4 prove that an emergent feature was found? What is the quantity on the x-axis in this figure?
2) Can you share the code of the architecture and its training or describe it in more detail?
3) Would you expect this method to find emergent features in standard, well-understood systems, such as Ising models?
4) Can you explicitly write down the prediction-only objective in Fig.2?

Overall, I find the approach promising but not yet tested on enough examples to support its ability to find emergent features.

**Limitations:**

There is no open access to the code.

---

> ### Author Rebuttal · Authors · 2024-08-07
>
> Many thanks for the feedback. We apologise for the lack of clarity in some of our explanations. We will improve these and add pseudo-code in the camera-ready version of the paper.
>
> Regarding the concern about not having enough examples, as we describe in more detail below and in the global rebuttal, we have conducted several additional benchmarks and evaluations. These new analyses include:
>
> 1. Two new real-world datasets, showing our method works well in both cases.
> 2. One more synthetic dataset, Conway’s Game of Life, which is arguably one of the most well-known examples of emergent behaviour. Since in this system there is a notion of what the emergent feature should be (known in the literature as “particles”), we can show that the learned feature captures the state of these particles.
> 3. Comparisons against baseline algorithms, making the case that our method finds features that standard algorithms based only on temporal predictions do not.
>
> In response to your questions:
>
> > How does Figure 4 prove that an emergent feature was found? What is the quantity on the x-axis in this figure?
>
> We apologise for the oversight in the labelling of this figure. The X-axis represents training time, and the Y-axes represent the emergence measure $\Psi$ and its adjusted variant $\Psi_A$. The plot proves that an emergent feature was found because the emergence measure is greater than zero, which is a sufficient condition for the feature being emergent (as shown by Rosas et al., 2020; Ref. [30] in the paper).
>
> > Can you share the code of the architecture and its training or describe it in more detail?
>
> Due to the constraints of the double-blind review we have been unable to share a Github link, but we will include it in the camera-ready version of the paper. Also, we apologise for the lack of clarity in our description – we will improve the description of the method and add a pseudo-code algorithm to the paper in the camera ready version.
>
> > Would you expect this method to find emergent features in standard, well-understood systems, such as Ising models?
>
> Although the behaviour of the Ising model is often described as emergent, our method is not readily applicable to the Ising model. This is because our method is explicitly designed for time series data (as the title indicates), but the Ising model doesn’t have time per se – the Ising model specifies a probability distribution over an ensemble of spin configurations, without any dynamics. There are of course several options to add dynamics to the spins (e.g. Metropolis-Hastings, Glauber, Kawasaki, etc.), but none of these are an integral part of the canonical Ising model. For this reason, if one were to choose one of these dynamics and apply our method, then the kind of conclusions one may draw is whether or not e.g. Glauber dynamics display emergent features, and not about whether the Ising model does.
>
> For this reason, we have chosen instead a different well-understood system that has dynamics and is widely considered to display emergence: Conway's Game of Life. As mentioned above, we show that our method can successfully find emergent features from Game of Life time series and that these features capture the state of the particles in the system.
>
> > Can you explicitly write down the prediction-only objective in Fig.2?
>
> The prediction-only objective is $I^\text{S}_\varphi(f\_{\theta}(X_t); f\_{\theta}(X\_{t+1}))$, i.e. the mutual information between the past and future of the learned representation. This quantity depends on the parameters of the critic $\varphi$ and the parameters of the representation learner $\theta$, and is calculated using the SMILE estimator as defined in Eq. (4).
>
> Finally, in response to the limitation:
>
> > There is no open access to the code.
>
> As mentioned above, we didn’t include a link due to the constraints of the double-blind review. We will add a link to a publicly accessible Github repository with all the code once the requirements of double-blind review are lifted. In the meantime, we have sent the AC an anonymised link to the code for the model architecture.

---

> > ### Comment · Reviewer_7mVS · 2024-08-09
> >
> > Thank you for your reply and clarifying my question regarding the notation.
> >
> > On the overall improved and clearer presentation for the final version I will put my trust in the new github repository, and also in the judgement in some of the reviewers who already found the first version easier to read than I did. They seem more familiar with the previous work that your method is based on.
> >
> > The new results, in particular using the game of life example, investigate exactly the kind of system I found missing as a benchmark in the original version of the article. I am going to raise the score in response.

---

### Official Review · Reviewer_PMxp · 2024-07-14

**Soundness:** 3
**Presentation:** 4
**Contribution:** 2
**Rating:** 6
**Confidence:** 4

**Summary:**

This paper introduces a method for learning the causally emergent representation of time series data. Based on the Partial Information Decomposition (PID) and ΦID definition of emergent variables, the paper utilizes variational information lower bounds to estimate and optimize the emergence objective function. The paper further includes a Minimum Mutual Information term and a penalty term, to reduce redundancy and discover diverse emergent variables, respectively. Experiments on a synthetic dataset and a primate brain activity dataset show that the method is able to discover diverse causally emergent representation.

**Strengths:**

Discovering causally emergent representations is a very interesting topic, and has significance in a wide range of scientific disciplines. The paper is inspirational and written clearly. Although the components of the method, i.e. definition of emergence objective function, and variational bounds for mutual information, are not new, their combination to discover causally emergent representations in a learnable way is interesting, and to my knowledge, novel.

**Weaknesses:**

As discussed above, the novelty is a little limited. This can be compensated by solid evaluations with a wide range of interesting datasets. I think the place the paper needs most improvements is more diverse and extensive evaluations. The paper can benefit from a few more datasets, both synthetic and real world, including the other datasets used in [1] and other references. If there exists baselines for discovering causally emergent representations, those baselines should also be compared against.



Reference:

1. Rosas, Fernando E., et al. "Reconciling emergences: An information-theoretic approach to identify causal emergence in multivariate data." PLoS computational biology 16.12 (2020): e1008289.

**Questions:**

N/A

**Limitations:**

The authors did not explicitly state the limitation of the paper.

---

> ### Author Rebuttal · Authors · 2024-08-07
>
> As mentioned in the overall rebuttal, we have now conducted a wider range of evaluations, both using the same architecture on more datasets and using other architectures on the same datasets.
>
> * Specifically, we add two more brain activity datasets which capture different aspects of neural dynamics: fMRI, with high spatial resolution and low temporal resolution; and MEG, with low spatial resolution but high temporal resolution. Our method is able to find emergent features in both cases.
>
> * We ran experiments on one more synthetic dataset used in Rosas et al. 2020, Conway's Game of Life. Here we found that our method can learn emergent features, and that these can be interpreted as encoding the state of the particles, as one would expect.
>
> * We compare against standard RNN and MLP architectures that predict the future state of the time series, without any information-theoretic loss function. We confirmed that neither the RNN or the MLP are able to learn emergent features.
>
> To the best of our knowledge, there are no established baselines for discovering causally emergent representations – if anything, the Game of Life is one of the most long-standing canonical examples of emergent behaviour generated by relatively simple rules, and we now show our method works on it. We would of course be happy to consider any other suggestions.
>
> Finally, in response to the limitation:
>
> > The authors did not explicitly state the limitation of the paper.
>
> We apologise for the lack of clarity on this regard. We state some of the limitations of our work in Section 4, although we acknowledge this needs to be more clearly laid out. For the camera-ready version we will add an explicit discussion of our method's limitations.

---

> > ### Comment · Reviewer_PMxp · 2024-08-12
> >
> > Thanks for the response. The added experiments provide more solid support for the claims of the paper. Thus, I increased my score.

---

### Official Review · Reviewer_aLEM · 2024-07-19

**Soundness:** 3
**Presentation:** 2
**Contribution:** 3
**Rating:** 7
**Confidence:** 3

**Summary:**

The paper presents a method for identifying emergent variables in time series data through a novel machine learning architecture. It uses unsupervised learning for representation and information theory to find emergent properties in systems, which are often complex and not easily describable at the microscale level. The paper is motivated by the fact that unsupervised learning can be a powerful tool for identifying emergent properties, but current approaches limit to only information theory.

The method rests on maximizing an objective defined by subtracting the mutual information of state variables at time t and the coarse graining at time t + 1 from the mutual information of the coarse grainings at t and t + 1. In other words, the amount of emergent information. Information theoretic definition of emergence is thus used to facilitate unsupervised learning. The method is tested on synthetic data and real-world ECoG brain activity data, demonstrating its ability to identify known and novel emergent features effectively.

Experiments are conducted on a synthetic dataset and a macaque brain activity dataset. For the synthetic dataset, the method is able to estimate the ground-truth value of psi (the difference that is central to the objective function). For ECoG data, skip connections were introduced into the architecture, and once again found emergent representations.

The paper concludes with a discussion on related (info theory) work, limitations, and future steps.

**Strengths:**

### Clarity
- Diagram features are well designed and results features are clear and salient
- Though writing is somewhat unstructured, the shorter-range explanations are well-done
- Methodology is given in detail. Lots of helpful explanation of relevant information theory, as well as the overall approach

### Quality
- Good to have primate brain data, though more interpretation would help
- Covers all the basic needs for a new method: real data, novel setup, suitable metrics (though they need more explanation)
- Experimental setup is well-designed to demonstrate that emergent variables are being learned

### Originality
- As far as I know, applying the information theoretic definition of emergent variables as an objective and training in this setting is novel

### Significance
- An innovative idea that shows promise. While there could be more experimentation, this is a promising and new direction.

**Weaknesses:**

### Clarity
- It's not immediately clear how to interpret results. The paper shows figures, but it doesn't explain them much. Interpreting them requires a lot of re-reading the methods section
- Writing is somewhat verbose and unstructured, and occasionally reads like a process statement

### Quality
- This idea is compelling and innovative! The loss built on MI of coarse grainings and state variables is intuitive while creating a solid foundation for taking advantage of the capabilities of unsupervised learning
- On the ECoG dataset, giving intuition/semantic understanding of emergent features (or at least attempting interpretation) would be cool
- Limited experiments on real data in general - ultimately, only one experimental setting is shown as far as I understand. The synthetic problem, while useful, is simple
- Lacking baselines or extensive comparison to existing methods, even if purely information-theoretic

### Significance
- It would help to have clearer comparison to existing methods so that we could see the value-add of this innovation, not just the novelty and value alone

**Questions:**

None

**Limitations:**

Yes

---

> ### Author Rebuttal · Authors · 2024-08-07
>
> In response to the specific weaknesses identified:
>
> > It’s not immediately clear how to interpret results. The paper shows figures, but it doesn’t explain them much. Interpreting them requires a lot of re-reading the methods section
>
> We apologise for the lack of clarity in our explanations. For the camera-ready version we will make sure the figures are explained in more detail and the text flows more smoothly.
>
> > On the ECoG dataset, giving intuition/semantic understanding of emergent features (or at least attempting interpretation) would be cool
>
> We enthusiastically agree with the reviewer that obtaining a semantic understanding of the learned emergent features in ECoG would be definitely cool. However, we see this as a longer-term research programme that our method is enabling. Since we do not have a sense of “ground truth” in terms of what the emergent features of brain activity are, this is difficult to do as part of this paper.
>
> In cases where we do have a reasonable sense of what these features might be (e.g. in the synthetic dataset and in Conway’s Game of Life), we conducted post-hoc analyses to interpret these features, and found that they align with our intuitions. For the camera-ready version we will emphasise the need for future work interpreting the learned emergent features in real-world data.
>
> > Limited experiments on real data in general - ultimately, only one experimental setting is shown as far as I understand. The synthetic problem, while useful, is simple
>
> As mentioned in the global rebuttal, we have now added experiments on two more real-world datasets, and demonstrate our method can successfully learn emergent features on both. We also add one more synthetic,but far less simple, dataset using Conway’s Game of Life system, and use our method to learn emergent features from it and interpret them with post-hoc analyses.
>
> > Lacking baselines or extensive comparison to existing methods, even if purely information-theoretic
>
> In addition to new ablation studies, showing comparisons against other information-theoretic loss functions, we now compare our method against a standard RNN and a standard MLP architecture (without any information theory). The results suggest that RNNs and MLPs do not learn emergent features on their own, supporting the value of our method.
>
> > It would help to have clearer comparison to existing methods so that we could see the value-add of this innovation, not just the novelty and value alone
>
> We agree that comparisons against existing methods can make a clearer argument in favour of our paper. As mentioned above, we now show that standard RNN or MLP do not spontaneously learn emergent features. Additionally, we show in post-hoc analyses that the combination of the learned emergent feature of our method and an RNN representation enables better predictions than either of them in isolation (informally: a small RNN plus an emergent feature is better than a large RNN), making a direct case for the value-add of our method.

---

> > ### Comment · Reviewer_aLEM · 2024-08-13
> > **Thanks for the response**
> >
> > Thanks for the response and the new work. I do think the new experiments help, and the writing looks better. In retrospect, I think I would have originally given a 5 or 6, and raised it to a 7, so I'm going to keep the current score.

---

### Official Review · Reviewer_pckS · 2024-07-19

**Soundness:** 3
**Presentation:** 4
**Contribution:** 3
**Rating:** 7
**Confidence:** 4

**Summary:**

The paper introduces a novel objective function and deep learning architecture that are targeted to extract emergent latent representations from time series data. Motivation is very clear. The definition of emergent latent representation interesting and useful. The utilization of mutual information estimators (lower bounds thereof) is smart. Evaluations are restricted to a fully artificial and a fully neurobiological dataset.

**Strengths:**

The study of emergence and its conceptual and mathematical formalization is of general interest to neural information processing and the involved (part-of) cognitive science subdiscipline within. The authors utilize an existing definition thereof [30] as well as an approximation technique of a lower bound on mutual information (SMILE, [32]), which they combine in a highly elegant manner to yield their learning architecture.

The usage of a linear temporal predictor with learnable residuals is a great way to bootstrap the system’s abilities.

Even multiple emergent latents can be successfully extracted.

A real-world dataset indicates applicability beyond artificial data.

Paper is very well written – relatively easy to comprehend and all steps taken are very well motivated and sufficient background is provided.

**Weaknesses:**

System evaluations are minimalistic and not as convincing as I had hoped. Both, comparisons to potential baseline algorithms as well as more ablations are missing.

Furthermore, one artificial dataset and one not well-motivated real-world neural dataset seems not enough to warrant publication.

In particular, I would have expected at least one if not multiple DL/Neural Network baselines that do not pursue the information-theoretic objective but simply a standard temporal prediction objective. Those probably do not work on the parity problem at all but at least an attempt seems needed. That is, use a DREAMER-like world model learning architecture with probabilistic latents and see if structure emerges.

Ablations could have explored more than just the same architecture without the penalty / adjustment term or without the macroscopic estimator. Further, in the artificial dataset the correlation coefficients $\gamma$ are quite high – was this necessary? When does this break down? Evaluations with a non-linear prediction pipeline would also be useful.

**Questions:**

Emergence comes in many forms – I wonder if you could discuss alternative definitions / alternative perspectives on the matter?

Line 52 – a “not” too much.

Eq (2): can you motivated the adjustment term further the summand (n-1)min_i… ? Are there alternatives to this that would be more targeted towards actually identifying true redundant mutual information?

Eq (4). Could you motivate the clip operator slightly more?

Line 102 should read “accurately”

Paragraph 113ff: maximize / minimize information – I am not sure if this is worded the right-way round – could you double check and slightly reword?

Line 137 – should not read “also”

Line 142 – one “is” too much

Line 190ff. removing the macroscopic MI term seems not to save much – or does it? The observation is interesting, but I wonder if the authors want to make a computational argument here as well.

About the biological dataset – this is very ad-hoc somewhat. What does this analysis tell us really except that there are some complex spatio-temporal statistical correlations in the data? I find this one marginally useful.

**Limitations:**

Unclear how robust this is as ablations and comparisons are not very extensive.

---

> ### Author Rebuttal · Authors · 2024-08-07
>
> As mentioned in the global rebuttal, we have now conducted extensive additional evaluations, including:
>
> * Two new real-world datasets and one new synthetic dataset;
> * New analyses on the existing synthetic dataset with lower correlation coefficients; and
> * Comparisons against baseline algorithms based only on standard temporal prediction.
>
> Overall, our method was able to learn emergent features in all these new scenarios, the resulting features were interpretable for the synthetic cases where a notion of “ground truth” emergence is known, and baseline algorithms (standard RNN and MLP) do not learn any emergent features.
>
> In response to your questions, we have fixed all the typos and rephrased the relevant sentences in the paper. Here are
> responses to the rest of the questions:
>
> > Emergence comes in many forms – I wonder if you could discuss alternative definitions / alternative perspectives on the matter?
>
> We totally agree that emergence comes in many forms, and we acknowledge that our paper could do a better job at placing our definition within the broader literature on emergence. For the camera-ready version we will include a discussion regarding alternative definitions of emergence, and how this could be explored in future work.
>
> > Eq (2): can you motivated the adjustment term further the summand $(n − 1) \min_i ...$ ? Are there alternatives to this that would be more targeted towards actually identifying true redundant mutual information?
>
> The reviewer is right in that the discussion surrounding the adjustment term could be expanded. For the camera-ready version we will include a proof of where this adjustment term comes from, and a discussion including what redundant information is and what role it plays in the formula. As a matter of fact, there are multiple measures of redundant mutual information in the literature, and a comparison in this context would be an interesting avenue for future work. However, we would like to emphasise that the current form of the adjustment is a recognised and well-defined measure of redundant information (in the sense that it satisfies many of the natural properties one would require of such a measure), as shown in the following paper:
>
> Barrett, Adam (2015). Exploration of synergistic and redundant information sharing in static and dynamical Gaussian
> systems. *Phys. Rev. E* 91, 052802. doi: [10.1103/PhysRevE.91.052802](https://www.doi.org/10.1103/PhysRevE.91.052802)
>
> > Line 190ff. removing the macroscopic MI term seems not to save much – or does it? The observation is interesting, but I wonder if the authors want to make a computational argument here as well.
>
> We agree, thanks for this observation. To clarify, there are in fact two arguments for doing this: one is the computational argument to save some compute; and the other is that in some cases we observed it stabilised learning dynamics (our hypothesis is that it alleviates the adversarial relationship between the different components of our objective). For the camera-ready version we will clarify these issues and include both arguments.
>
> > About the biological dataset – this is very ad-hoc somewhat. What does this analysis tell us really except that there are some complex spatio-temporal statistical correlations in the data? I find this one marginally useful.
>
> Our motivation to study emergence (as described in the introduction) is to understand how cognitive systems encode information in macroscopic variables, which naturally motivates us to test our methods on data from biological brains. The fact that the same method that successfully identifies particles in the Game of Life detects emergent dynamics in brain data is extremely encouraging, suggesting that neural systems may encode information into “glider-like” collective features. This paper provides direct empirical evidence supporting this idea. Moreover, it is particularly encouraging that the emergent character is observed in multiple neuroimaging modalities that cover multiple spatial and temporal scales of the brain's dynamics.
>
> Needless to say, there is much future work to be done examining precisely what the role of emergent dynamics in the brain is, and how these emergent features contribute to brain function. We see this as a longer-term research programme, which our method is enabling for the first time. Since we do not have a sense of “ground truth” of what the emergent features of brain activity should be, this is difficult to do as part of this paper (although note that for cases where such ground truth is known, such as the bitstrings and the Game of Life, the results align with our intuitions). For the camera-ready version, we will emphasise the significance of these findings and the need for future work interpreting the learned emergent features in real-world data.
>
> Finally, in response to the limitation:
>
> > Unclear how robust this is as ablations and comparisons are not very extensive.
>
> As mentioned in the global rebuttal, we have now performed:
> 1. Further ablation analyses, showing the expected behaviour that key properties of the architecture (e.g. ability to learn diverse features) are lost if the corresponding elements of the objective are removed; and
> 2. Further comparisons with other, non-information-based architectures (e.g. RNNs, MLPs).

---

> > ### Comment · Reviewer_pckS · 2024-08-14
> >
> > Great rebuttal with new datasets and results as well as very good answers!
> > The provided additional information make me confident that the final paper will turn out to be very good and very useful for the ML community - at least for those that are interested in emergence. Please make sure that you use the additional page left accordingly (submission was less than 8 pages long).
> > I raise my score by one (i.e., to 7 - accept).

---

### Author Rebuttal · Authors · 2024-08-07

We would like to thank all the reviewers for their time and thoughtful comments on our paper. We are encouraged by the positive feedback, and are thankful for the constructive suggestions that have let us identify and address several limitations of our paper.

We have added responses to each reviewer’s specific comments, and provided answers to all of the reviewers’ questions (where applicable). More broadly, we have identified a number of overarching themes that multiple reviewers brought up and would like to address here. Below is a non-exhaustive list of work we have done motivated by the reviewers’ suggestions, together with a description of changes we are ready to make for the camera-ready version of the paper:

* **Evaluations in more real-world datasets**. We have conducted experiments on two more real-world datasets involving two different brain scanning modalities: functional magnetic resonance imaging (fMRI) and magnetoencephalography (MEG). Results reveal that our method was able to successfully learn emergent features in both datasets (rebuttal Fig. 1). This shows that our method is flexible and effective; and it provides a stronger motivation for the real-world data analyses as showing emergent behaviour is relevant across multiple scales of brain
activity – microscale with ECoG, mesoscale with MEG, and macroscale with fMRI.

* **Evaluation in a new synthetic dataset and additional ablation analyses**. We have also used our method to successfully find emergent features in Conway’s Game of Life (a prominent and long-standing example of emergent behaviour), which complements our results via a second synthetic dataset from a well-understood system (rebuttal Fig. 2). In addition, we have also conducted further ablation studies, which clarify the specific role of each component of our architecture. Finally, we also replicated our results for additional $\gamma$ values in our synthetic dataset, which provide further evidence confirming our previous findings (rebuttal Fig. 3).

* **Better interpretation of learned features**. Although reverse-engineering the function of emergent features of the brain is a larger, long-term research problem, we are able to provide clear interpretations of learned emergent features in systems where some notion of “ground truth” is available. Specifically:
  1. For the “random bits” synthetic dataset, we show that the learned feature can be interpreted as capturing the parity of the bitstring (paper Fig. 2).
  2. For the new results on the Game of Life, we show that the learned feature captures the state of the so-called particles (i.e. gliders) in the system (rebuttal Fig. 2).

* **Comparison against standard architectures**. We have compared the performance of our method against a standard RNN and a standard MLP trained only on a temporal prediction objective, both on real-world (rebuttal Fig. 1) and synthetic data (rebuttal Fig. 4). The results show that neither the RNN nor the MLP by themselves learn the emergent features of the data, and thus our method can provide a unique value for predicting datasets with emergent dynamics. As quantitative evidence for this claim, we show that the combination of a learned emergent feature and the RNN representation can predict the future state of the synthetic dataset better than either of them in isolation, highlighting the value-add of our method in conjunction with other algorithms.

* **Improved presentation**. In the camera-ready version we will improve the description of our architecture and our figures, provide a pseudo-code algorithm, and discuss our framework in the broader context of alternative definitions of emergence.

* **Code availability**. We will add a link to a publicly available Github repository once the requirements of the double-blind review are lifted. In the meantime, we have sent the AC an anonymised link to the code for the model architecture.

We hope this new work addresses some of the reviewers’ concerns, particularly regarding the addition of a broader range of evaluations on new real-world and synthetic datasets. Please do not hesitate to ask more questions if any aspect of our work remains unclear. And, once again, we would like to thank all the reviewers for their extremely valuable feedback.

---

### Decision · Program_Chairs · 2024-09-25

**Decision:**

Accept (poster)

**Comment:**

This work attempts to combine insights from the literature on emergence with differentiable estimators of mutual information to look for emergent features in time series data. The authors explore the presence of emergent features in some synthetic data sets and in data from neuroscience experiments.

Reviewers appreciated the problem formulation and the combination of ideas from information decomposition with differentiable estimators. Reviewers also found the problem statement clear and of interest to the NeurIPS community. Additional experiments performed by the authors in response to reviews were also judged to substantially strengthen the paper. One weakness noted by all reviewers was the difficulty of interpreting what emergent features were learned in the real data, as the finding is based on the estimated value of $\Psi$, without any particular insight as to what that implies.

On balance, all reviewers considered this a valuable contribution, and I concur. Accept.